# Facile Synthesis of Novel Disperse Dyes for Dyeing Polyester Fabrics: Demonstrating Their Potential Biological Activities

**DOI:** 10.3390/polym14193966

**Published:** 2022-09-22

**Authors:** Alya M. Al-Etaibi, Morsy Ahmed El-Apasery

**Affiliations:** 1Natural Science Department, College of Health Science, Public Authority for Applied Education and Training, Fayha 72853, Kuwait; 2Dyeing, Printing and Textile Auxiliaries Department, Textile Research and Technology Institute, National Research Centre, 33 El Buhouth St., Dokki, Cairo 12622, Egypt

**Keywords:** disperse dyes, polyester fabrics, microwave irradiation, ultraviolet protection factor

## Abstract

Original work showed the composition of the dyes and the antimicrobial/UV protective properties of a series of dyes obtained in our laboratories over the past twelve years in an easy way using microwave technology and their comparisons with conventional methods. The results we obtained clearly indicated that by using the microwave strategy, we were able to synthesize the new disperse dyes in minutes and with a much higher productivity when compared to the traditional methods, which took a much longer time, sometimes up to hours. We also introduced ultrasonic technology in dyeing polyester fabrics at 80 °C for an environmentally friendly approach, which was an alternative to traditional dyeing methods at 100 °C; we obtained a much higher color depth than traditional dyeing methods reaching 102.9%. We presented both the biological activity of the prepared new dyes and the fastness properties and clearly indicated that these dyes possess biological activity and high fastness properties.We presented through the results that when dyeing polyester fabrics with some selected disperse dyes, the color strength of polyester fabrics dyed at high temperatures was greater than the color strength of polyester fabrics dyed at low temperatures by 144%, 186%, 265% and 309%. Finally, we presented that a ZnO or TiO_2_ NPs post-dyeing treatment of polyester fabrics is promising strategy for producing polyester fabrics possess multifunction like self-cleaning property, high light fastness, antimicrobial and anti-ultraviolet properties.

## 1. Introduction

Since time immemorial, colors have played a major role in human life, as they are deeply linked to their means of living, their traditions, their ideas, their customs and their concepts. Since the dawn of humanity, man has been fascinated by the diversity of colors; therefore, he has been fascinated by beauty of their many colors, which has had far-reaching effects in his religious, emotional and social life. Azo disperse dyes were the most widely used synthetic dyes of the past decade because of their ease of manufacture and wide application in cosmetics, textile dyeing and printing. It is worth noting that heterogeneous, bright-colored. Azo dyes are widely used for dyeing polyester [1,2,3,4,5,6,7,8,9,10,11,12]. Polyester is one of the polymers containing ester bonds obtained from the polycondensation reaction of diol and dicarboxylic acid (Figure 1) [4].

Polyester (PET) is the most hydrophobic fiber among ordinary fibers, and this fiber is less prone to wrinkling and has excellent washability and abrasion resistance. It is known that the polyester dyeing process is performed by water dyeing using dispersed dyes in the presence of high temperatures and pressures. It is known that azo dyes are widely used in textile dyeing in view of environmental pollution, so azo dyes harm the environment if their effluents are not treated. Among these treatments are modern techniques using ultrasound, infrared, microwaves, and ultraviolet rays to improve the fabric’s absorption capacity for dyes and reduce effluents load from dyeing processes [13,14,15,16,17,18,19,20,21,22,23,24]. Some recent reports [1,2] have used ultrasound energy to improve the dyeing of polyester fabrics when dyeing with disperse dyes by modifying the surface of the fabric. Additionally, some studies [3] confirmed that UV rays can be used to improve the color fastness properties without compromising the chemical properties of polyester fabric. One of the environmentally friendly advantages of using microwave energy is the rapid heating to high temperatures that allows for greater ease of reactions, because increasing the frequency of molecular vibrations during microwave irradiation speeds up these reactions as well as reduces the use of solvents [25,26,27,28,29,30,31,32,33,34,35,36,37,38,39,40,41,42,43,44,45,46,47,48,49,50,51,52,53,54,55,56,57,58,59,60,61,62,63,64,65,66]. Skin cancer and sensitive skin are primarily caused by prolonged exposure to ultraviolet radiation from sunlight. Therefore, the best choice is to locate well-designed clothing composed of ultraviolet-blocking materials. One of the qualities cloth should have is the ability to filter ultraviolet rays, either by containing an ultraviolet absorber or by being dyed, as dyed fabrics offer better sun protection than bleached materials. Therefore, it could be claimed that dyed cloths with darker colors may provide better protection from the Sun rays [1,4]. According to a literature survey, pyrazolopyrimidines, pyridones and enaminones are significant key intermediates in the production of new azo dyes, particularly disperse dyes with numerous biological applications. Because of their facile synthesis, low cost, and great efficacy against bacterial infections, they are now widely employed as antimicrobial agents [4]. In general, the kinds of substituted function present in the heterocyclic ring structure, as well as the form of the heterocyclic system, have a significant impact on the antibacterial activity of the entire molecule. The majority of the matching substituted analogues had stronger antibacterial inhibitory activities that were on par with or better than the reference drugs. A lot of researchers are now interested in creating smart fabrics for human wellbeing. The need to create materials that improve defense against microorganisms such as bacteria or fungi is expanding. Due to their propensity to retain moisture, fabrics can act as transporters and ideal environments for the growth of germs such as bacteria, especially when they come into contact with the human body. Therefore, it is essential to create textiles that are microbe-resistant. To produce smart fabrics, we had to improve the functional performance of polyester fabrics by using nanotechnology. To increase their anti-bacterial and UV-protective properties and to achieve this goal, nanoparticles were used, especially metal oxides and metal nanoparticles. The potential mechanism behind titanium dioxide NPs’ antibacterial action is their photocatalytic function, which results in the generation of extraordinarily active oxygen species, such as super oxide anions, hydrogen peroxide, singlet oxygen, and hydroxyl radicals, which kill bacteria. Due to the presence of the outer cell wall membrane in the G-ve bacteria, which acts as a barrier to the antibacterial action, the NPs-treated polyester fabric demonstrated greater antibacterial activity against G+ve bacteria in companion with G-ve bacteria. In this review article, we present the contributions of our laboratories over the past twelve years using modern technologies such as ultrasound or microwave technology to synthesize many new disperse dyes while demonstrating their biological activities. We will not only present the use of modern technologies such as ultrasound or microwave technology in dyeing polyester fabrics and compare them with traditional dyeing methods but also discuss the process of treating polyester fabrics dyed with ZnO or TiO_2_ NPs and whether it achieved a UV protection factor, as well as the ability of these polyester fabrics to improve the properties of self-cleaning, light fastness and, antimicrobial activities.

## 2. Materials

It is true that microwave and/or ultrasound technologies have many uses and unique advantages. However, we should also note that with a better understanding of the physical principles underlying the coupling mechanisms between microwave radiation or ultrasound energy and matter, we may be able to expand the applications of these technologies to cutting-edge scientific applications. In 2011, we formed a research team with the intention of synthesizing novel disperse dyes utilizing microwave heating by reacting hydrazine hydrate **4** with hydrazonocyanoacetate 3 [58]. By combining ethyl cyanoacetate **1** and diazoniun salt **2**, hydrazone **3** was formed (Figure 1) [58,59]. As a result, dye 5 was formed when hydrazone **3** and hydrazine hydrate 4 interacted in ethanol. We found that compound **5** quickly condensed with acetyl acetone **6** to produce the disperse dye 7 under microwave irradiation (c.f. Figure 1, Figure 2). Compound **5** interacted with enaminones 8a-d through microwave irradiation to create the disperse dyes **9a–d** (c.f. Figure 1, Figure 3).

In 2012, we were able to explain the one-pot synthesis of compound **7** with a better outcome using microwave irradiation, using hydrazone 3, hydrazine hydrate **4** and acetyl acetone **6**. According to Figure 1 [58], disperse dye 7 can exist in isomeric forms. We also reported in 2012 [58] that compounds **8a–d**, hydrazone **3**, and hydrazine hydrate **4** could be easily reacted as one-pot synthesis to synthesize compounds **9a–d** (c.f. Figure 1).

One of the sequences used in the synthesis of the disperse dye **16a–h** was the coupling of malononitrile **10** with diazonium salt **2** to generate compound **12**. (Figure 2). An increase in the aryl proton signal resulted from irradiating the hydroxyl signal, according to the Nuclear Overhauser Effect (NOE) measurements [59].

In order to create the disperse dye **14**, hydrazine hydrate **4** refluxed with compound **12** [60]. When compound **11** was combined with diazonium salt **2** in the presence of ethyl alcohol/sodium acetate, a high yield of hydrazone **13**—85%, was produced. In order to help construct the structure of **13**, NOE experiments were conducted. The results demonstrated that irradiating the NH signal at 12.1 ppm induced an amplification of the aryl proton resonances at 7.39 and 6.80 ppm (Figure 2) [60].

Hydrazone **13** and hydrazine hydrate were combined to create the disperse dye **15** by refluxing the mixture for four hours while adding ethanol as a solvent. According to NOE difference measurements, irradiation of the NH signal at 11.94 ppm corresponding to compound **15** boosted the methyl proton signal at 2.36 ppm. Through refluxing for an hour in the presence of acetic acid and sodium acetate, dyes **14** or **15** easily condensed with the enaminones **8a–d** to generate disperse dyes **16a–d** or **16f–h** (Figure 2) [60].

It is well known that dye **22** was created by combining different types of diazonium chloride **21** with diketones **23** to create substituted arylazodiketones **24**, which may then be condensed with cyanoacetamide derivatives **25** using either a traditional heating method or a microwave heating method [67,68,69,70,71]. (Figure 3).

On the other hand, the first reaction method for the synthesis of these azo dyes utilized the interaction between diazonium chloride and pyridones. We described a three-component condensation of methyl propionylacetate **17** as -ketoesters, ethyl cyanoacetates **1** and ethyl amines **18** to generate pyridine **19** in 2014 [61] by utilizing microwave irradiation at 160 °C for 20 min. It is important to note that in 2013 [57], we created compound **19** by heating it traditionally for 6 h. Two tautomeric forms, **19** and **20**, of this compound readily equilibrated in solution (Figure 3).

Based on X-ray crystallographic structure determination, pyridine **19** could be combined with different diazonium salts **21** to create the disperse dyes **22a–i** that exist in the hydrazone tautomeric form, as shown in Figure 3. (Figure 4 and Figure 5).

In 2015 [72,73,74], we were able to produce novel disperse dyes that were safe for the environment. We accomplished this by reacting enaminones **8e** and **8f** with the phenyl diazonium salt **21a** in an acidic medium to produce the disperse dyes **26a** and **26b** that were 3-oxo-2-(phenylhydrazono)-3-p-arylpropionaldehydes (Figure 4).

Due to the significance of thiophene molecules and their potent biological activity, numerous researchers have investigated aminothiophene derivatives as azo-disperse dyes in dyeing synthetic fibers [75]. We are aware of no reports of the corresponding arylazothiopyridazines as prospective monochromatic disperse dyes, despite the numerous studies on the efficacy of these compounds in the production of dyes. In 2014, we synthesized some arylazothienopyridazines **28a–d** using a straightforward and environmentally friendly conventional method by reacting 7-Amino-4-benzotriazol-1-yl-2-p-tolyl-2H-thieno[3,4-d]pyridazin-1-one **27** with various diazonium salts **21**. This work continued the growing interest in the synthesis of arylazothienopyridazinones (Figure 4) [75].

## 3. Dyeing

The disperse dyes **5**, **7**, **9a–d**, **14**, **15**, and **16a–h** were used to dye polyester fabrics with hues 2% using microwave heating at 130 °C, producing a range of color shades (Table 1).

Color strength was assessed at the maximum wavelength λ_max_ and given as K/S values. The Kubelka–Munk equation was used to perform K/S [1,21].
$$K/S=\frac{{\left(1-R\right)}^{2}}{2R}-\frac{{\left(1-R_{0}\right)}^{2}}{2R_{0}}$$
where R is the decimal fraction of the reflectance of the dyed fabric; R_0_ is the decimal fraction of the reflectance of the not dyed fabric; K is the absorption coefficient; and S is the scattering coefficient.

The dyes have a good affinity for polyester materials at the stated temperatures of 130 °C, as shown by the color strength values K/S listed in Table 1, giving it brilliant hues. In order to treat the dyeing baths, we decided to reuse the dye bath from the previous study and extend the dyeing time from 60 to 90 min without adding any new dye. It is difficult for the human eye to distinguish the dye’s color constancy after the dye bath was reused.

Additionally, it is evident from the results that the color strength K/S values for the initial dyeing process were 2.12, 3.79, 5.95, 5.81, 4.64 and 4.73, and the K/S values for the subsequent dyeing bath process were 1.67, 3.56, 5.47, 3.49, 3.54 and 3.95. After comparing the K/S values of the two methods, we discovered that the K/S for the second dyeing process had reuse rates of 78.77%, 93.93%, 91.93%, 60.06%, 6.29 and 83.5%. Table 1 makes it very evident that dye **16d** yields far stronger colors than dyes **16a–c** and **16h**. Polyester fabrics were dyed using compounds **22a–i**, **26a**, **26b** and **28a–d** at a high pressure, high temperature and 2% shade.

The materials were colored in a variety of hues, from yellow to violet. Following that, the fastness properties of polyester fabrics were used to evaluate the dyeing features of such materials. According to the K/S estimations in Table 1, the **22a–i** hues exhibited a significant attraction for polyester materials, and all color strengths were typically positive.

The International Commission on Illumination (CIE) established the CIELAB (Color space) psychometric coordinates in 1976, where *L** stands for lightness and (*C*) for chroma. The information in Table 2 shows that almost all colored polyester fabrics conveyed a similar hue when the dye’s hue was expressed as (*h*) values. The positive estimates of *b** showed that the dyed polyester materials’color hues shifted in a reddish direction [61].

The total color difference ∆*E* was measured by using an UltraScan Pro (Hunter Lab, USA) 10° observer with D65 illuminant, d/2 viewing geometry and measurement area of 2 mm. The total color difference ∆*E** between the sample and the standard was calculated using the following equation:$$\Delta E*=\sqrt{{\left(\Delta L^{*}\right)}^{2}+{\left(\Delta a^{*}\right)}^{2}+{\left(\Delta b^{*}\right)}^{2}}$$
where ∆*L**, ∆*a** and ∆*b** are the derivatives of corresponding parameters, respectively

The hues **22e–g** were lighter and brighter than the **22b–d** because electron-donating groups were included into the benzene ring, which decreased brightness, while electron withdrawing groups improved lightness and brightness.

### Dye Uptake

The colorimetric parameter values obtained for the high temperature and low temperature colored polyesters are listed in Table 2, which demonstrates that the high temperature dyed fabrics were darker than the low temperature dyed fabrics. Color strength (K/S) showed the dye uptake. The K/S values for dyes **22h** and **22i** were 19.38, 12.63, and 4.74 and 3.46, respectively, for materials colored at high and low temperatures. These results show that the dye uptake of fabrics colored at high temperatures was greater than that of materials dyed at low temperatures by 309% and 265%, respectively.

Additionally, the K/S values for dyes **26a** and **26b** were 17.59, 16.69, and 12.21 and 8.97, respectively, for materials colored at high and low temperatures. These results show that the dye uptake of the materials dyed at high temperatures was more than that of low temperature dyeing by 144% and 186%, respectively. According to the results, high temperature dyeing is an environmentally safe technique since it can lessen the pollution load in colored dye effluents, which would otherwise have a negative impact on the environment.

Furthermore, as the rate of dye penetration into the filament increased, the pressure increased along with the dyed fabrics at high temperatures. The kinetic energy of the dye molecules may be increased by the temperature, and the temperature may also cause the polyester fibers to inflate, increasing the dyeing rate relative to low temperature dyeing.

According to Table 2′s findings, the K/S value for dyeing by ultrasound at 80 °C is 9.07, while for dyeing by the traditional method at 100 °C, it is 4.47. This means that the first approach using ultrasound is superior to the traditional method by 102.9% for the disperse dye **22h** [1].

## 4. Fastness Properties

According to the tests of the American Association of Textile Chemists and Colorists [58], the fastness characteristics of the dyed samples were evaluated against perspiration, washing and light. The information in Table 3 demonstrates that the color fastness characteristics of polyester textiles dyed with dyes **5**, **7**, **9a–d**, **14**, **15**, **16a–h**, **22a–i**, **26a–b** and **28a–d** were measured.

The results for dyed polyester fabrics dyed with dyes **5**, **7**, and **9a–d** are shown in Table 3, where the ratings for color fastness to light and washing were good and very good and the rating for color fastness to perspiration was excellent [60].

Data obtained by testing the color fastness characteristics of polyester fabrics colored with dyes **14**, **15**, and **16a–h** are shown in Table 3. With the exception of dye **16d**, all of the tested dyes had outstanding washing and perspiration fastness, according to the fastness values listed in Table 3. 

The polyester colored materials’ light fastness showed moderate fastness. The type of substituents in the diazonium component had a considerable impact on the light fastness. As a result, this moiety’s fading rate should increase for electron-donating groups, whereas it should decrease for electron-withdrawing groups.

The results (Table 3) show that the inclusion of a methyl group in dyes **16b** and **16g** produced a reduction in light fastness to 3; our idea is consistent with these findings. The chlorine atom in the dyes **16c** and **16h**, on the other hand, was linked to an increase in light fastness to 4 and 6, respectively [61].

Data acquired by assessing the color fastness characteristics of polyester fabrics dyed with dyes **22a–i** are displayed in Table 3 as fastness data. The grades for fastness are listed in Table 3, which demonstrates very good levels of fastness for perspiration and outstanding levels of fastness for washing. The polyester colored fabrics’ moderate fastness is shown by their light fastness.

The type of substituents in the diazonium component has a considerable impact on the light fastness. The addition of electron-withdrawing (bromine, chorine or nitro) substituents increase the dye **22e**, **22f** and **22g**’s light fastness to (3–4), (3–4) and (5), respectively. In terms of the following factors, the results generally indicated that the dyed fabric might have good fastness: (i) Effective dye molecule diffusion throughout fabric fibers. (ii) The dye molecule’s size was regarded as being fairly large. (iii) The solubility and detergency of fabric dyeing were unaffected by any solubilizing groups [61].

## 5. In Vitro Cytotoxicity Screening

One of the most important markers for biological evaluation in in vitro investigations is cytotoxicity. Different cytotoxicity mechanisms exist for compounds in vitro, such as the suppression of protein synthesis or irreversible binding to receptors [64]. The synthetic disperse dyes **22h** and **22i** were tested for their preliminary anticancer activity against four human cell lines, including HepG-2 cells (for the treatment of hepatocellular carcinoma), MCF-7 cells (for the treatment of breast cancer), HCT-116 cells (for the treatment of colon cancer) and A-549 cells (for the treatment of lung carcinoma). The IC_50_ values, the concentration needed to prevent 50% of the development of the culture when the cells were exposed to the tested disperse dyes for 48 h, were calculated using various concentrations of the two disperse dyes. Disperse dye **22h** displayed significant activity, as shown by Table 4 and Figure 6, Figure 7, Figure 8 and Figure 9, with IC_50_ values of 23.4, 62.2, 28 and 53.6 g/mL in HePG-2, MCF-7, HCT-116 and A-549 cells, respectively. In contrast, dispersion dye **22i** showed negligible activity in HePG-2, MCF-7, HCT-116 and A-549 cells with IC_50_ values of 196, 482, 242 and 456 g/mL, respectively.

## 6. Antioxidant Activity (DPPH Radical Scavenging Activity)

In vitro testing of the two dispersion dyes’ antioxidant properties used their capacity to scavenge DPPH free radicals. The IC_50_ value of the dyes—the dose needed to suppress the production of DPPH radicals by 50%, was used to describe their antioxidant activity. Data from Table 4 show that disperse dye **22h** had a moderate antioxidant activity and outperformed ascorbic acid as the standard, which had an IC_50_ of 14.2, while disperse dye **22i** had a poor antioxidant activity with an IC_50_ of 191.6. (Figure 10).

## 7. Antimicrobial Activities of Dyes 3, 5, 7, 9a–d, 22a–i and 26a,b

The agar well diffusion method was used to investigate the antibacterial effects of the produced dyes **3**, **5**, **7** and **9a–d** against various bacteria and yeast. The information in Table 5 reveals encouragingly effective antibacterial activity. While the other disperse dyes exhibited moderate to poor antibacterial properties, disperse dyes **3** and **5** had strong antibacterial activity against Gram-positive bacteria.

All of the substances tested inhibited the development of *Candidia albicans* after six days of incubation. Additionally, the agar well diffusion method was used to test the dispersion dyes **22a–i** and **26a,b** for their antibacterial properties against a range of bacteria and yeast.

In addition, Figure 11 shows that *Candida albicans* re-grew in the formed zone surrounding the wells containing compound **3**. This may reflect the cytostatic effect of the chemicals rather than their cytolytic effects. Note that the plate color changed with the increase in the incubation time indicates a complete diffusion of the chemical used in the agar as the incubation period increased.

**Figure 11 polymers-14-03966-f011:**
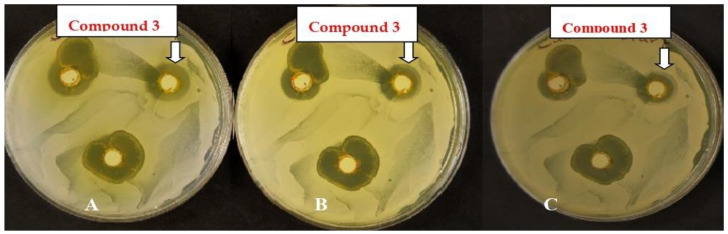
*Candida albicans* treated with 10mg ml^−1^ of compound **3** after one day (**A**), three days (**B**) and six days (**C**) of incubation. It is worth noting that Figure 10 shows the cytolyticeffects of dyes number 5, 7 and 9a on *Candida albican* where after one, three and six days of incubation, the inhibition zone did not change (Figure 12).

While, the cytostatic effect of the same dyes is clear on the plates inoculated with *Bacillus subtilus.* The growth of *B. subtilus* resumed in the inhibited area after six days of incubation, which may indicate that as the concentration/toxicity of dyes number **5**, **7** and **9a** reduce due to possible evaporation of these dyes or diffusion in the media, the effect of these dyes on *B. subtilus* decrease, and the organisms start growing again (Figure 13).

Based on data for the inhibition zone diameter for the dispersion dyes **22a–g**, Table 5 shows that all of the tested dyes demonstrated strong positive antibacterial activities against the studied pathogens. For all of the examined bacteria, disperse dye **22a** had a cytolytic effect, with no growth being seen in the inhibition zone.

## 8. UV Protective Properties of Untreatedand Treated Polyester Fabrics with ZnO or TiO_2_ Nanoparticles NPs

The UV protection factor (UPF) was calculated to obtain ultraviolet-protective qualities. UPF is a characteristic that endows materials such as polyester textiles with ultraviolet protection properties. Table 6 shows UV blocking data for polyester fabrics treated with ZnO or TiO_2_ nanoparticles. According to the UPF data in Table 6, treated polyester fabrics with ZnO nanoparticles had UPF values of 173.25 for disperse dye **26a** and 190.59 for disperse dye **26b**, respectively. This shows that treated polyester fabrics with dye **26a** have UPF values that are significantly higher than treated polyester fabrics with dye **26b**.Table 6 further demonstrates that treated polyester fabrics have higher UPF values than untreated polyester fabrics, with respective values of 141.88 for dye **26a** and 122.37 for dye **26b**. Additionally, the UPF data in Table 6 shows that treated polyester fabrics with TiO_2_ nanoparticles have UPF values of 283.60 for disperse dye **22h** and 34.9 for disperse dye **22i**, respectively. This shows that treated polyester fabrics with dye **22h** have much higher UPF values than treated polyester fabrics with dye **22i**. Table 6 further demonstrates that treated polyester fabrics have higher UPF values than untreated polyester fabrics, with values of 236.2 for dye **22h** and 25.5 for dye **22i**, respectively.

## 9. Self-Cleaning of Untreatedand Treated Polyester Fabrics with ZnO or TiO_2_ NPs

One of the benefits of polyester fabrics coated with nanoparticles is the conversion of absorbed light into self-cleaning substances to remove stains. To gain the self-cleaning properties of nano ZnO or TiO_2_ particles, the photodegradation of methyl red and methylene blue adsorbed on nano ZnO or TiO_2_ treated polyester fabrics was explored (Table 7).

After 24 and 12 h of UV exposure, Table 7 displays the results of methyl red and methylene blue stains on polyester fabrics treated with ZnO or TiO_2_ NPs. For polyester fabrics treated with ZnO or TiO_2_ NPs, a partial discoloration of methyl red and methylene blue stains brought on by ultraviolet radiation was seen. When polyester fabric is treated with ZnO or TiO_2_ NPs, thin layers of ZnO or TiO_2_ nanoparticles develop, increasing the fabric’s hydrophobic qualities. A hydrophobic surface stops dirt from adhering, keeping the polyester surface clean at all times. The results showed that the highest rates of photodegradation on the surface were 60–70% for methyl red stains treated with ZnO NPs after 24 h and 60–80% after 12 h for methylene blue stains treated with TiO_2_ NPs (Table 7).

## 10. Light Fastness of Untreated and Treated Polyester Fabrics with ZnO or TiO_2_ NPs

The light fastness property of all colored polyester fabric samples that were treated with ZnO or TiO_2_ NPs of disperse dyes **26a**, **26b**, **22h**, and **22i** were measured, and the results were more significant. Table 7 shows that, with the exception of dye **26a**, using ZnO or TiO_2_ NPs more effectively resulted in treated polyester fabrics having stronger light fastness than the untreated samples (Table 7).

## 11. Antimicrobial Activity of Untreated and Treated Polyester Fabrics with ZnO or TiO_2_ NPs

The untreated and treated polyester fabrics are tested against the pathogenic fungi *Aspergillus flavus* and *Penicillium chrysogenum,* as well as the Gram-positive bacteria *Bacillus subtilis* and the Gram-negative bacteria *Klebsiella pneumoniae*. Table 8 revealed that untreated polyester fabrics with ZnO or TiO_2_ NPs did not exhibit antibacterial activity against all of the microorganisms [64,73].

While only *Bacillus subtilis* was the target of the antibacterial activity of nano ZnO treated polyester fabrics of dispersion dye **26a**, nano ZnO treated polyester fabrics of disperse dye **26b** were effective against both *Bacillus subtilis* and *Klebsiella pneumoniae*. ZnO NPs had antibacterial activity against bacteria, and its mechanism involved either ZnO nanoparticles or TiO_2_ nanoparticles influencing bacterial membranes to inhibit bacterial growth. Alternatively, nano ZnO may enable the generation of peroxide, which may offer antibacterial properties [65,73].

According to the antifungal screening results given in Table 8, the two harmful fungi *Aspergillus flavus* and *Penicillium chrysogenum* were not resistant to the treated polyester dyed fabrics with TiO_2_ NPs of dispersion dye **22i**. *Aspergillus flavus* and *Penicillium chrysogenum* were resistant to *Aspergillus flavus* and the treated polyester colored fabrics with TiO_2_ NPs of dispersion dye **22h**.

## 12. Conclusions

Our original work highlighted that the synthesis of new disperse dyes, with the help of microwaves, could be carried out in the presence of small quantities or sometimes in the absence of any amounts of organic solvents, which are harmful to the environment, in a short time that did not exceed minutes and gave great yields compared to the synthesis of those dyes using traditional methods. It is known that cytotoxicity is one of the most important markers of biological evaluation in laboratory tests, so we presented two examples of new disperse dyes as one of the examples of in vitro cytotoxicity examination.We showed that these dyes possess anticancer activities against some common cancers such as lung, breast, liver and colon cancer. The added value of these new disperse dyes was also discussed and presented by showing the biological activity and clarifying that these dyes have a great biological activity against Gram-positive and Gram-negative bacteria, as well as various fungi and yeasts. It is worth noting here that polyester fabrics dyed with these dyes had biological activity, which makes these fabrics able to be used in many medical activities. We presented and discussed the use of ultrasound energy to dye polyester fabrics, as it gave a high color strength compared to traditional dyeing methods. In this review, we also discussed the methods of treating polyester fabrics with nano-zinc oxide or nano-titanium dioxide and presented the advantages of this strategy in endowing the dyed polyester fabrics with multiple functions, such as self-cleaning property, maximizing light fastness property and maximizing antimicrobial activities.

## Data Availability

Not applicable.

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
