# Peer review of "Facile Synthesis of Novel Disperse Dyes for Dyeing Polyester Fabrics: Demonstrating Their Potential Biological Activities"

_polymers, 2022, doi:10.3390/polym14193966_

Round 1
Reviewer 1 Report
In this work authors present a review on dyes synthesis and antimicrobial / UV protection characteristicss of a series of dyes obtained in authors’ labs. It is a mixed review-original paper which requires some substantial changes:
- please focus on your original work, i.e. antimicrobial / UV protection of selected dyes, and delete the sub-section 2. „Synthesis and Characteristics”; instead introduce „Materials” sub-section with the structures of the dyes and references to your previous works for readers interested in their synthesis,
- details of the polyester fabrics dying procedure should be presented,
- please change the title to read e.g. „Facile Synthesis of Novel Disperse Dyes for Dyeing Polyester fabrics: Demonstrating their Potential Biological Activities”,
- Please re-write the abstract – it should be more informative,
- Introduction needs to be expanded to cover the antimicrobial / UV protection mechanisms for various dyes,
- 5. „In Vitro Cytotoxicity Screening” – please provide images for cytotoxicity screening,
- „Self-cleaning of untreated and treated polyester fabrics with ZnO or TiO2 NPs.” – please add NPs characteristics and describe fabrication route of polyester fabrics with ZnO or TiO2. What about the agglomeration effects?
- 12. „Conclusions” have to be changed / completed.
Moreover, English needs to carefully checked.
Reviewer 2 Report
This paper give us useful imformation. However, this paper needs some amendment.
1. The purpose of this paper is too vague. The authors should describe the novelty or importance, and originality of this paper in the abstract.
In particular, it should be described that the research results using ultrasonic or microwave technology are superior to conventional heating methods in terms of eco-friendliness, physical properties, biocompatibility, and antibacterial properties.
2. In this paper, In this paper, the authors describe compounds synthesized using microwaves. Then, each chapter describes the properties of each compound synthesized, making it difficult to read.Therefore, it will be easier to read and understand this paper if authors group similar techniques or materials to organize each chapter and describe the properties, strengths, and problems of the synthesized compounds.
3. In my opinion, it would be better to remove 'our laboratories' in the title.
Round 2
Reviewer 1 Report
The revised version of this work - although improved - still requires some amendments. Authors are requested to address these issues:
- Introduction needs to be expanded to cover the antimicrobial / UV protection mechanisms for various dyes,
-in "Abstract" and "Conclusions" please change "In this review, we present a review on..." into "original work".
Author Response
We do thank the respective reviewer. Please find the attached file

Reviewer 2 Report
The authors changed the manuscript according to the suggestions and answered quite clearly to the questions. Overall this version is improved in comparison to the first one, therefore I recommend publication in the present form.
Author Response
Thank you very much for your great efforts.